

# Associations between pro-inflammatory cytokines and fatigue in pregnant women

Haiou Xia[1], Xiaoxiao Zhu[1] and Chunxiang Zhu[2]

[1] School of Nursing, Fudan University, Shanghai, China
[2] Obstetrical Ward, Obstetrics & Gynecology Hospital of Fudan University, Shanghai, China

## ABSTRACT

**Background**. Fatigue is one of the most prevalent symptoms among pregnant women. In patients with various diseases, pro-inflammatory cytokines are associated with fatigue; however, such associations are unknown in pregnant women.

**Objectives**. The objective of this study was to examine the associations between pro-inflammatory cytokines and prenatal fatigue.

**Methods**. A cross-sectional study was conducted on 271 pregnant Chinese women in their third trimester of pregnancy. Patient-reported Outcome Measurement Information System (PROMIS) was used to evaluate women's prenatal fatigue. Using enzyme-linked immunosorbent assay (ELISA), the serum concentrations of four pro-inflammatory cytokines, including tumor necrosis factor alpha (TNF-$\alpha$), interleukin 1 beta (IL-1$\beta$), interleukin 6 (IL-6) and interleukin 8 (IL-8), were measured. The data was analyzed by correlation analysis and general linear regression analysis.

**Results**. In this sample, the mean (standard deviation) of fatigue scores was 51.94 (10.79). TNF-$\alpha$ ($r = 0.21$, $p < 0.001$), IL-6 ($r = 0.134$, $p = 0.027$) and IL-8 ($r = 0.209$, $p = 0.001$) were positively correlated to prenatal fatigue, although IL-1$\beta$ was not. TNF-$\alpha$ ($\beta = 0.263$, $p < 0.001$), along with sleep quality ($\beta = 0.27$, $p < 0.001$) and depression ($\beta = 0.376$, $p < 0.001$) independently predicted prenatal fatigue.

**Conclusions**. TNF-$\alpha$ was identified as an independent biomarker for prenatal fatigue in our study. Reducing pro-inflammatory cytokines may be a unique method for lowering prenatal fatigue and, consequently, enhancing mother and child health.

## INTRODUCTIONS

Fatigue is a subjective sensation of persistent and overwhelming exhaustion or tiredness that reduces functioning capacity (*Pugh et al., 1999*). Prenatal fatigue is one of the most often reported complaints from pregnant women (*Cheng & Pickler, 2014*). According to longitudinal studies, prenatal fatigue is most prevalent and severe in the third trimester of pregnancy (*Chien & Ko, 2004*; *Milligan & Pugh, 1994*). In the third trimester of pregnancy, prenatal fatigue was related with adverse clinical outcomes, including prenatal hospitalizations (*Luke et al., 1999*), delivery mode (*Chien & Ko, 2004*) and preterm birth (*Stinson & Lee, 2003*). Additionally, it may contribute to postnatal depression by

Corresponding author
Haiou Xia, hsxia@fudan.edu.cn

impairing postnatal fatigue and impairing baby development by decreasing maternal-fetal attachment (*Bakker et al., 2014*; *Mason, Briggs & Silver, 2011*). Thus, expanding our understanding of prenatal fatigue in the third trimester of pregnancy may benefit both mothers and their children.

Mounting evidence suggest pro-inflammatory cytokines could protect the mother and fetus from infections in healthy pregnancy (*Leff-Gelman et al., 2016*). Pro-inflammatory cytokines are quantified in peripheral blood by immunological markers such as tumor necrosis factor alpha (TNF-$\alpha$), interleukin-1beta(IL-1$\beta$), interleukin-6 (IL-6) and interleukin-8 (IL-8). Concentrations of TNF-$\alpha$ were found to be elevated in peripheral blood across pregnancy (*Kraus et al., 2010*; *Blackmore et al., 2011*), whereas those of IL-1$\beta$ could decrease gradually (*Ferguson et al., 2014*). IL-6 and IL-8 levels decreased during the first and second trimesters but rose from the second to third (*Ross et al., 2016*; *Stokkeland et al., 2019*). As a result, the third trimester of pregnancy may contain more pro-inflammatory cytokines than the first two trimester (*Mor & Cardenas, 2010*). Excessive inflammation, on the other hand, were linked to unfavorable clinical outcomes during pregnancy, including miscarriage (*Christiansen, Nielsen & Kolte, 2006*), pre-eclampsia (*Maher et al., 2019*) and preterm birth (*Gilman-Sachs et al., 2018*). This was consistent with the most prevalence and severity of prenatal fatigue in the third trimester (*Chien & Ko, 2004*; *Milligan & Pugh, 1994*). Therefore, pro-inflammatory cytokines may have a role in prenatal fatigue.

Pro-inflammatory cytokines have been implicated with physiological indicators of fatigue in a variety of individuals, including those with chronic fatigue syndrome and pulmonary disease (*Al-Rawaf, Alghadir & Gabr, 2019*; *Al-Shair et al., 2011*). The function of pro-inflammatory cytokines in prenatal fatigue, on the other hand, is much less understood. Furthermore, no association between pro-inflammatory cytokines and prenatal fatigue have been identified. Thus, further research is needed to explore the relationship between pro-inflammatory cytokines and prenatal fatigue.

Investigating the relationships between pro-inflammatory cytokines and prenatal fatigue may contribute to our understanding of prenatal fatigue. Additionally, targeting pro-inflammatory cytokines may be an unique technique for reducing prenatal fatigue and thereby improving mother and child clinical outcomes. The purpose of this study was to investigate the associations between pro-inflammatory cytokines (TNF-$\alpha$, IL-1$\beta$, IL-6, and IL-8) and prenatal fatigue in women's third trimester of pregnancy.

## METHODS

### Participants

From June to September 2021, a cross-sectional study was done in a public hospital in Shanghai, China. This study enrolled a sequential sample of 271 people. Women who met the following criteria were included: (a) were in their third trimester of pregnancy; (b) were at least 20 years old; and (c) could read and write Chinese. Exclusion criteria included the following: (a) prior diagnosis of chronic fatigue or depression; (b) significant health problems such as heart failure; (c) twins or multiples; and (d) inability to do blood tests. The work was ethically approved by the institutional research ethics committees at School

of Nursing in Fudan University (IRB#2019-12-06) and the Obstetrics and Gynecology Hospital affiliated to Fudan University (No.2020-191). Additionally, each participant signed an informed consent form.

## Procedure

The researcher identified and recruited eligible participants when women waited for their usual obstetric assessment, which included a blood test. While awaiting the blood test, participants were invited to complete self-report questionnaires. All questionnaires were cross-checked on-site. Demographic data and clinical data about gestational weeks were extracted from hospital medical records. Only 1.0–1.5 milliliter (ml) of fresh serum was required to determine pro-inflammatory cytokine concentrations in this investigation. To avoid collecting further peripheral blood, this study examined the residual fresh serum from blood samples within three days following the blood test under aseptic circumstances in the hospital laboratory. Residual fresh serum was collected and stored at $-80\ ^\circ$C until pro-inflammatory cytokines were measured.

## Measurements

Demographic and clinical covariates, independent variables, and one dependent variable were included in the descriptive data. Age (mean (standard deviation), M(SD)), prenatal body mass index (BMI, M(SD)), parity (primipara = 1, multipara = 0, number(percent)), employment (employed = 1, unemployed = 0, $n$ (percent)), educational level (<college = 1, college or bachelor = 2, >bachelor = 3, $n$ (percent)), and hometown (Shanghai = 1, other places = 0, $n$ (percent)) were included as demographic covariates. Covariates in the clinical setting included gestational weeks (M(SD)), depression (M(SD)), and sleep quality (M(SD)).

Depression was measured by the Edinburgh Postnatal Depression Scale (EPDS). The EPDS, a ten-item questionnaire, is one of the most extensively used self-reported measures for assessing depression in pregnant women (*Cox, Holden & Sagovsky, 1987*). Each item on the scale is coded from 0 to 3 (0 = never, 1 = sometimes, 2 = often, 3 = always). As a result, the raw sum score spans between 0 and 30. The global scale's internal consistency (0.76) and short-term test-retest reliability (0.98) are both satisfactory (*Guedeney & Fermanian, 1998*).

Sleep quality was measured by the Pittsburgh Sleep Quality Index (PSQI). It was developed in 1989 by Daniel J. Buysse and was primarily used to measure respondents' subjective sleep quality over the preceding month (*Buysse et al., 1989*). In a sample of college students, *Liu et al. (1995)* translated the scale into Chinese. The PSQI comprised of 19 items that were self-evaluated and five items that were rated by others. Each factor is scored on a scale of 0 to 3, and the total PSQI score is the sum of the values for each item. Five more evaluation items are not scored. The total PSQI score is between 0 and 21, with higher scores indicating decreased sleep quality. It had a high degree of internal consistency (*Mollayeva et al., 2016*).

Four pro-inflammatory cytokines were used as independent variables (TNF-$\alpha$, IL-1$\beta$, IL-6, and IL-8). TNF-$\alpha$, IL-1$\beta$, IL-6, and IL-8 serum concentrations were determined
using an enzyme-linked immunosorbent assay (ELISA): (a) Take out a 96-well plate for Elisa and Dilute 10x Coating Buffer solution to 1x Coating Buffer solution with PBS. The first antibody of the tested protein is diluted with 1x Coating Buffer solution, and added 100 ul to each well, stored in 4 °C and coated overnight. (b) Each well is washed with 300 ul 1x PBST solution containing 0.05% Tween20 for five times. (c) 100 ul 1x Dilluent liquid is added to each well and sealed for 1 h at room temperature. (d) Each well is washed for five times with 1x 0.05% PBST liquid. (e) Add standard samples and samples to a 96-well plate with a volume of 100 ul per well and incubated for 2-4 h at room temperature. (f) Each well is washed for five times with 1x 0.05% PBST liquid. (g) Dilute Avidin-HRP with 1x Dilluent liquid, add 100 ul to each well, and incubate at room temperature for 30 min. (h) Each well is washed for seven times with 1x 0.05% PBST liquid. (i) Avoid light, add 100 ul TMB to each well, and observe the color change in the well. The color development time is usually 5–30 min. When the standard sample is fully colored, 50 ul 2 mol/L dilute sulfuric acid is immediately added to each well to stop the color reaction. (j) Use enzyme labeling instrument test OD450/570 nm. Additionally, the unit of measurement for four pro-inflammatory cytokines was ng/ml.

Fatigue was the sole dependent variable in this study. Fatigue was assessed using the National Institutes of Health's (NIH)-developed eight-item short form of the Patient-reported Outcome Measurement Information System (PROMIS Fatigue SFs 8a, abbreviated as PROMIS in this article) (*Cella et al., 2010*). Each of the eight items of PROMIS had a five-point rating from 1–5 (1 = not at all, 2 = a little, 3 = moderately, 4 = mostly, 5 = completely). The raw sum score was the sum of the scores of the 8 items. The final score was derived using the raw sum score of the T-score metric, which had a mean of 50 and a standard deviation of 10; higher scores indicated more fatigue (*Cella et al., 2010*). Cronbach's alpha and intra-class correlation coefficients for the PROMIS were reported to be 0.929 or more (*Kamudoni et al., 2021*). In addition, the PROMIS website (https://www.healthmeasures.net/score-and-interpret/interpret-scores/promis/promis-score-cut-points) listed <55 scores as normal, 55~60 scores as mild, 60~70 scores as moderate and >70 scores as severe fatigue. In this study, we used a cut-off point of 55 on the PROMIS for fatigue and considered pregnant women with a score of at least 55 to be fatigued. The incidence of prenatal fatigue was calculated by dividing the number of fatigued women by the total number of study participants and multiplying the result by 100 percent.

## Statistical analysis

In this study, SPSS version 23.0 (IBM, Armonk, NY, USA) was used for the data analysis. Data analysis included descriptive analysis, Pearson analysis, Spearman analysis and general linear regression analysis. Descriptive analysis were used for all variables, including demographic and clinical covariates, independent variables, and dependent variable (prenatal fatigue). For prenatal fatigue in this study, PROMIS scores were used. Continuous data are represented by the mean (standard deviation), and categorical data by the number of cases (percentage).

The objective of this study was to analyze the associations between the four pro-inflammatory cytokines and prenatal fatigue. First, correlation analysis was used to analyze correlations between all covariates and prenatal fatigue. Of them, Pearson analysis was for continuous (*e.g.*, age) and Spearman analysis for categorical (*e.g.*, parity) variables. Second, Pearson analysis was used to examine the correlations between pro-inflammatory cytokines (TNF-$\alpha$, IL-1$\beta$, IL-6, and IL-8) and prenatal fatigue. Variables with $p < 0.05$ were initially considered to be significantly related to prenatal fatigue and then introduced into final regression model. Additionally, general linear regression analysis was used to further characterize the associations between pro-inflammatory cytokines and prenatal fatigue. We used the forward selection method to construct models. The final model included pro-inflammatory cytokines associated with prenatal fatigue as independent variables, fatigue scores as the dependent variable, and all covariates related to prenatal fatigue as covariates. In the regression model, variables with a significance level of $p < 0.05$ were considered to be predictors of prenatal fatigue.

## RESULTS

### Descriptive data

In this study, participants were pregnant women with an average gestational week of 35.55 (SD = 1.36) weeks, age of 30.15 (3.97) years, and prenatal BMI of 21.28 (2.71). Other demographics of the 271 participants were also shown in Table 1. Additionally, these participants reported depression with an EPDS score of 5.0 (3.52) and sleep quality with a PSQI score of 6.94 (3.15). These demographic and clinical variables could be covariates.

Four pro-inflammatory cytokines, including TNF-$\alpha$, IL-1$\beta$, IL-6 and IL-8, were independent variables in this study. In Table 1, the average concentrations of these cytokines were also listed. As TNF-$\alpha$, IL-1$\beta$, IL-6 and IL-8 were not normally distributed, their quartiles were also provided: 45.48 (26.12, 74.08), 14.52 (7.32, 38.96), 118.28 (48.19, 237.90) and 764.12 (474.45, 1409.93) ng/ml, respectively.

In addition, the dependent variable of this study was fatigue. The mean (SD) of prenatal fatigue was 51.94 (10.79) scores. We set the cut-off score for PROMIS on fatigue at 55. Consequently, 92 of the 271 women in this sample reported at least mild fatigue during their third trimester of pregnancy. The prevalence of prenatal fatigue was therefore 33.9% in this study.

### Correlations between covariates and prenatal fatigue

Utilizing correlation analysis, correlations between demographic and clinical variables and prenatal fatigue were analyzed (Table 2). Pearson analysis was for continuous (*e.g.*, age) and Spearman analysis for categorical (*e.g.*, parity) variables. As a result, hometown (Shanghai = 1, other places = 0) was negatively related to prenatal fatigue ($r = -0.12$, $p = 0.049$), which indicating that women from Shanghai could experience a higher level of prenatal fatigue compared to those from other places of China. However, gestational weeks ($r = 0.151$, $p = 0.013$), depression ($r = 0.49$, $p < 0.001$) and sleep quality ($r = 0.539$, $p < 0.001$) were positively related to prenatal fatigue, which indicating that women with

**Table 1  Descriptive data of participants ($N = 271$).**

| Variables | M (SD)/N (%) |
|---|---|
| Demographic covariates | |
| Age | 30.15 (3.97) |
| Prenatal BMI | 21.28 (2.71) |
| Parity | |
|     Primipara | 221 (81.5) |
|     Multipara | 50 (18.5) |
| Employment | |
|     Employed | 254 (93.7) |
|     Unemployed | 17 (6.3) |
| Educational level | |
|     <College | 63 (23.2) |
|     College or Bachelor | 150 (55.4) |
|     >Bachelor | 58 (21.4) |
| Hometown | |
|     Shanghai | 83 (30.6) |
|     Other places | 188 (69.4) |
| Clinical covariates | |
| Gestational weeks | 35.55 (1.36) |
| Depression (EPDS), scores | 5.0 (3.52) |
| Sleep quality (PSQI), scores | 6.94 (3.15) |
| Independent variables | |
| TNF-$\alpha$, ng/ml | 58.59 (52.87) |
| IL-1$\beta$, ng/ml | 33.36 (42.99) |
| IL-6, ng/ml | 167.82 (173.22) |
| IL-8, ng/ml | 926.8 (594.63) |
| Dependent variable | |
| Fatigue (PROMIS), scores | 51.94 (10.79) |

**Notes.**

M, Mean; SD, standard deviation; N, number; %, percentage; BMI, Body Mass Index; TNF-$\alpha$, tumor necrosis factor alpha; IL-1$\beta$/6/8, Interleukin-1$\beta$/6/8.

larger gestational weeks, greater level of depression or worse sleep quality could have greater degree of prenatal fatigue.

## Correlations between pro-inflammatory cytokines and prenatal fatigue

Table 2 showed Spearman analysis to analyze correlations between pro-inflammatory cytokines (TNF-$\alpha$, IL-1$\beta$, IL-6 and IL-8) and prenatal fatigue. As a result, TNF-$\alpha$ ($r = 0.21$, $p < 0.001$), IL-6 ($r = 0.134$, $p = 0.027$) and IL-8 ($r = 0.209$, $p = 0.001$) were positively correlated to prenatal fatigue. But IL-1$\beta$ ($r = 0.032$, $p = 0.599$) was not related to prenatal fatigue. This indicated that women with higher levels of TNF-$\alpha$, IL-6 or IL-8 could experience increased level of prenatal fatigue.

**Table 2  Correlations between other variables and prenatal fatigue ($N = 271$).** Correlation analysis was used to explore correlations between covariates, independent variables, and prenatal fatigue. A correlation coefficient of correlation analysis ($r$) with its $p$-value was used.

|  | r | p |
|---|---|---|
| Demographic covariates |  |  |
| Age | 0.01 | 0.87 |
| Prenatal BMI | 0.004 | 0.949 |
| Parity (primipara = 1, multipara = 0) | −0.069 | 0.255 |
| Employment (employed = 1, unemployed = 0) | −0.035 | 0.566 |
| Educational level (low = 1, middle = 2, high = 3) | −0.036 | 0.557 |
| Hometown (Shanghai = 1, other places = 0) | −0.12 | 0.049 |
| Clinical covariates |  |  |
| Gestational weeks | 0.151 | 0.013 |
| Depression (EPDS), scores | 0.49 | <0.001 |
| Sleep quality (PSQI), scores | 0.539 | <0.001 |
| Independent variables |  |  |
| TNF-$\alpha$, ng/ml | 0.21 | <0.001 |
| IL-1$\beta$, ng/ml | 0.032 | 0.599 |
| IL-6, ng/ml | 0.134 | 0.027 |
| IL-8, ng/ml | 0.209 | 0.001 |

**Notes.**

$r$, correlation coefficient of correlation analysis; BMI, Body Mass Index; low, <college; middle, college or bachelor; high, >bachelor; TNF-$\alpha$, tumor necrosis factor alpha; IL-1$\beta$/6/8, Interleukin-1$\beta$/6/8.

## Associations between pro-inflammatory cytokines and prenatal fatigue

Based on results from previous sections, a general linear regression model was built to explore the associations betweem pro-inflammatory cytokines (TNF-$\alpha$, IL-6 and IL-8) and prenatal fatigue (Table 3). In this model, TNF-$\alpha$, IL-6 and IL-8 was regarded as the independent variable and prenatal fatigue as dependent variable. And hometown, gestational weeks, depression and sleep quality were controlled as covariates. We used the forward selection method for model building, so Table 3 only showed results of variables remained in the final model. As a result, the model fit was satisfactory ($F = 68.19$, $p < 0.001$). TNF-$\alpha$ ($\beta = 0.263$, $p < 0.001$), along with sleep quality ($\beta = 0.27$, $p < 0.001$) and depression ($\beta = 0.376$, $p < 0.001$), explained 42.7% of the variance in prenatal fatigue in the model. This model indicated for every 1 ng/ml increased in TNF-$\alpha$, prenatal fatigue could increased by 0.263 scores, that was, TNF-$\alpha$ could predicted prental fatigue.

## DISCUSSIONS

To our knowledge, this is the first study to examine pro-inflammatory cytokines in peripheral serum in association to prenatal fatigue. Our findings validated independent fatigue biomarkers, providing additional evidence for therapies targeting pro-inflammatory cytokines to alleviate prenatal fatigue and improve mother and child health.

**Table 3** Associations between pro-inflammatory cytokines and prenatal fatigue ($N = 271$). A general linear regression model was built for associations between pro-inflammatory cytokines and prenatal fatigue. A standardized regression coefficient ($\beta$) with its $p$-value and 95% confidence interval were presented. TNF-$\alpha$ was identified as an independent predictor of prenatal fatigue.

| | $\beta^{\S}$ | SE | $\beta$ | $t$ | $p$ | $\beta^{\S}$ 95% CI |
|---|---|---|---|---|---|---|
| Constant | 36.032 | 1.284 | | 28.064 | <0.001 | [33.504, 38.560] |
| TNF-$\alpha$ | 0.054 | 0.01 | 0.263 | 5.581 | <0.001 | [0.035, 0.073] |
| Sleep quality (PQSI) | 0.925 | 0.178 | 0.27 | 5.191 | <0.001 | [0.574, 1.276] |
| Depression (EPDS) | 1.152 | 0.159 | 0.376 | 7.266 | <0.001 | [0.840, 1.465] |

**Notes.**
A general linear regression analysis.
$\beta^{\S}$, unstandardized regression coefficient; SE, standard estimate; $\beta$, standardized regression coefficient; $t$, statistics for $t$-test; $p$, $p$-value; $\beta^{\S}$ 95% CI, 95% confidence interval of $\beta^{\S}$; TNF-$\alpha$, tumor necrosis factor alpha; PSQI, pittsburgh sleep quality index; EPDS, edinburgh postnatal depression scale.

## Prevalence and severity of prenatal fatigue

Our study is the first to apply the PROMIS to assess prenatal fatigue in China, which provide new reference for the distribution of PROMIS scores among pregnant women for the NIH database.

Because the majority of fatigue scales in the existing literature lack a cut-off value, there are limited reports on the prevalence of prenatal fatigue during the third trimester (*Zhang et al., 2021a*). Prenatal fatigue was evaluated in a Swedish population using one question and revealed that almost 90% of women experienced prenatal fatigue, with 34% reporting fatigue practically daily and 58% reporting fatigue occasionally (*Rodriguez, Bohlin & Lindmark, 2001*). A total of 75% of participants in a different Chinese sample reported prenatal fatigue during the third trimester when asked if fatigue was an issue for themselves (*Cheng & Pickler, 2014*). Our sample's prevalence of prenatal fatigue was 33.9%, which was very close to the 34% (by a single question that if they experienced practically daily) but far lower than 75~90% (by a single question about whether fatigue was a problem for themselves). Thus, we found almost one third of pregnant women experienced at least mild fatigue during their third trimester of pregnancy. Our findings are crucial in the field of prenatal fatigue by providing its prevalence based on PROMIS, and future study should verify our findings based on scales with cut-off points.

The mean (SD) of fatigue in our sample was 51.94 (10.79) scores, which was higher than that in US general population (mean of 50 and SD of 10) (*Liu et al., 2010*). However, the mean (SD) of prenatal fatigue in our study was lower than those in other pregnant populations, where they were 55.25 (7.53) (*Alcantara et al., 2018*), 56.03(5.96) (*MoghaddamHosseini et al., 2021*) and 58.3(0.79) (*Lyon et al., 2014*). Despite the fact that we all focused on pregnant women, the characteristics of pregnant women in previous studies differed from ours (women with gestation week of 35.55 weeks). In particular, Joel's study focused on pregnant women across pregnancy (*Alcantara et al., 2018*), Debra's on women during their second trimester of pregnancy (*Lyon et al., 2014*) and Vahideh's on women with 36.98 gestational weeks (*MoghaddamHosseini et al., 2021*). Since the current reports on prenatal fatigue based on PROMIS are limited to the few

studies listed above, future study can examine the our findings in populations of various cultures.

## Associations between pro-inflammatory cytokines and prenatal fatigue

An earlier study established the connection between monocyte chemotactic protein(MCP, a component of inflammation) and prenatal fatigue (*Cheng & Pickler, 2014*). Inflammation was speculated to be associated with prenatal fatigue. However, the links between pro-inflammatory cytokines (as the main components of inflammation) and prenatal fatigue were still unknown. In our investigation, TNF-$\alpha$ was able to predict prenatal fatigue independently, demonstrating a role for pro-inflammatory cytokines in the pathogenesis of prenatal fatigue. Despite the paucity of publications on the associations between TNF-$\alpha$ and prenatal fatigue, this association has been commonly seen in other populations, such as patients with cancer (*Zhang et al., 2021b*), chronic fatigue syndrome (*Domingo et al., 2021*) and gastrointestinal disease (*Norlin et al., 2021*). So our findings firstly provided evidence suggesting TNF-$\alpha$ involvement in prenatal fatigue pathways. Future studies could verify our findings in pregnant women and make therapies targeting pro-inflammatory cytokines to alleviate prenatal fatigue.

Although IL-6 and IL-8 did not predict prenatal fatigue independently after adjusting for covariates, they were related to prenatal fatigue in Spearman analysis. A prior study demonstrated a substantial association between IL-6 and IL-8 and other prenatal stress, such as anxiety or depression (*Osborne et al., 2018*; *Cassidy-Bushrow et al., 2012*). Also, IL-6 and IL-8 were associated with fatigue in other populations. Particularly, in patients with advanced cancer or multiple sclerosis, IL-6 was the most significant cytokines associated with fatigue (*De Raaf et al., 2012*; *Malekzadeh et al., 2015*). And IL-8 may be a predictor of fatigue in post-injury patients (*Crichton et al., 2021*). Thus, IL-6 and IL-8 may also play a role in prenatal fatigue. In addition, correlation between IL-1$\beta$ and prenatal fatigue was not significantly. This could be because IL-1$\beta$ has a linear fall tendency across pregnancy and a relatively low level in the third trimester of pregnancy (*Ferguson et al., 2014*). Hence, associations between IL-6, IL-8, and IL-1$\beta$ and prenatal fatigue require validation, and further investigations into such associations should be performed.

## Limitations

Some limitations of this study are noteworthy. Firstly, participants are recruitment with consecutive sampling in an obstetrics and gynecology hospital in Shanghai, China. Thus the findings of this study cannot be generalized to all pregnant women worldwide. Second, pregnancy consists of three trimesters with varying levels of pro-inflammatory cytokines; however, our cross-sectional study only focused on the third trimester. This resulted in cross-sectional rather than longitudinal relationships between pro-inflammatory cytokines and prenatal fatigue. Longitudinal studies of such associations from the first to third trimester of pregnancy is required. Thirdly, this study assessed only four pro-inflammatory cytokines (TNF-$\alpha$, IL-1$\beta$, IL-6 and IL-8) rather than the whole pro-inflammatory cytokines profile. Therefore, our findings regarding the possible inflammatory physiology of prenatal fatigue were limited and needed replication in multi-center and longitudinal investigations with the entire pro-inflammatory cytokines profile.

## CONCLUSIONS

Despite increased interest in the function of pro-inflammatory cytokines in fatigue physiology, very few studies have investigated the associations between pro-inflammatory cytokines and prenatal fatigue. TNF-$\alpha$ was identified as an independent biomarker for pregnant fatigue in our study.

## ACKNOWLEDGEMENTS

We would like to extend our gratitude to all participants in this study, as well as the professionals who advised us on the project's design and implementation.

### Funding

This work was supported by the Nursing Scientific Research Fund of Fudan University (No. FNSF201903). The funders had no role in study design, data collection and analysis, decision to publish, or preparation of the manuscript.

### Grant Disclosures

The following grant information was disclosed by the authors:
Nursing Scientific Research Fund of Fudan University: FNSF201903.

### Competing Interests

The authors declare there are no competing interests.

### Author Contributions

- Haiou Xia conceived and designed the experiments, authored or reviewed drafts of the article, and approved the final draft.
- Xiaoxiao Zhu conceived and designed the experiments, performed the experiments, analyzed the data, prepared figures and/or tables, authored or reviewed drafts of the article, and approved the final draft.
- Chunxiang Zhu conceived and designed the experiments, performed the experiments, authored or reviewed drafts of the article, and approved the final draft.

### Human Ethics

The following information was supplied relating to ethical approvals (*i.e.*, approving body and any reference numbers):

This work was supported by the ethics committees of the Obstetrics and Gynecology Hospital affiliated to Fudan University (No.2020-191).

### Data Availability

The raw measurements are available in the Supplementary File.

## Supplemental Information

Supplemental information for this article can be found online at http://dx.doi.org/10.7717/peerj.13965#supplemental-information.

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
