# Peer review of "Associations between pro-inflammatory cytokines and fatigue in pregnant women"

_PeerJ, doi:10.7717/peerj.13965_

## Round 0.1 · original submission · Major Revisions

Thank you for your submission. Based on the reviewers' comments, I suggest major revisions for this manuscript. Please respond to reviewers' comments.

Reviewer 1 ·

Basic reporting

Clear and unambiguous, professional English is used throughout.

Experimental design

The research question is well defined, relevant & meaningful. It is stated how research fills an identified knowledge gap.

Validity of the findings

All underlying data have been provided; they are robust, statistically sound, & controlled.

Additional comments

General Comments

Reviewed is the manuscript “Associations between pro-inflammatory cytokines and fatigue in pregnant women” submitted by Haiou Xia, et, al. This manuscript uses a cross-sectional study to present the connections between pro-inflammatory cytokines and prenatal fatigue. The article is well organized with a smooth flow of information during the explanation of each method. It is well-written, with very few clerical errors, and the style and layout are very well articulated. Overall, the authors clearly demonstrate their approach and detail the performance gained in this research field and the article meets the required standards for publication after minor edits.

Specific Comments
- Consider adding 95% confidence intervals to Figures 1 – 4 to better illustrate the difference of independent variables between fatigue and non-fatigue groups.
- Add more details to the table 2 legend description.
- The title of table 2 is “Correlations between pro-inflammatory cytokines and prenatal fatigue (N=271).”. The authors could include correlation analysis like Pearson’s correlation or modify the title to fit its content.
- A related paper section is recommended, to further compared with other relevant results.
- Review the written portion of the thesis for grammatical errors, especially the subject-verb agreements.

Reviewer 2 ·

Basic reporting

Thank you for the invitation of reviewing the manuscript. The authors investigated the distribution and association of four pro-inflammatory cytokines levels and fatigues in pregnant women. The manuscript was generally well-written and the study is ethically conducted with IRB approval. Ultimately, I think this work is suitable for publication in PeerJ, following revisions related to cut-point of fatigue measurement, PROMIS, and other statistical inconsistency.

Experimental design

None.

Validity of the findings

None.

Additional comments

Major comments:
The major issues in the manuscript are the choice of PROMIS cut-off score and the lack of consistency between description of statistical methods and what were actually shown in tables/figures, plus some misuse of statistical terminology. They are listed below:
1. The choice of cut-off point PROMIS score needs more explanation and discussion. First, for the purpose of study, I don’t see any rationale for categorizing PROMIS score. If keep the PROMIS score as continuous and fit it into a multivariate linear model, the interpretation would still be easy and useful. Second, I think choice of cut-off score needs to be reconsidered. The authors choose the mean in reference group as their cut-off point, but people not necessarily have fatigue symptoms if they have a >50 score since the mean is just an average of population. Also, the wording fatigue/non-fatigue is too decisive. For example, I don’t believe people with score of 49 and 51 are significantly different from each other, but the way authors putting it make it sounds very different. Actually, the PROMIS website (https://www.healthmeasures.net/score-and-interpret/interpret-scores/promis/promis-score-cut-points) listed <55 as normal, 55-60 mild, 60-70 as moderate and >70 as severe fatigue in their reference group. If authors decided to keep the logistic model, please explain the benefit of categorizing PROMIS score and how cut-point was chosen.
2. Four pro-inflammatory cytokines and other variables like depression, sleep quality, and parity should be independent variables in this study since authors were trying to use them to predict fatigue. On the other hand, fatigue is the dependent variable for the logistic model because the predicted result of fatigue is dependent on the values of cytokines and other variables. In Table 1, line 168 – 170, and line 159, authors put fatigue as independent variable and the four cytokines as dependent variables.
3. For Table 1, I think it’s better to present all independent variables by fatigue/non-fatigue groups and add a p-value column since the Methods stated (line 152 - 154) comparison between two fatigue groups but I don’t see any comparisons on variables except for cytokines.
4. In line 182 – 183, authors stated that four cytokines were fitted into logistic models, but the Methods part said the model only included variables that are significant different between fatigue groups (TNF-α, IL-6, and IL-8). Please clarify this. Also, my understanding is that all cytokines were put into one model but there is only TNF-α in Table 2. Please mention why other cytokines were omitted.
5. In Table 1, it’s unclear to me that why some continuous variables were presented in the format of mean(SD) while some were presented as median(p25, p75). Also, the statistical analysis in Methods didn’t mention any use of medians and quantiles. I would suggest using only one format across all continuous variables in Table 1 and make the language and table consistent.
Minor comments:
1. Line 61 – 63: It looks like these two sentences are speaking the same thing as TNF-α and ILs are pro-inflammatory cytokines.
2. Line 155: What do you mean by saying “positive” here?
3. Line 155 – 156: The meaning of “binary logistic regression (forward)” is unclear to me. Do you mean that you used forward selection method for model building?
4. For Figure 1-4, what values are presented? Mean or median? I would also suggest adding a notation if significantly different.
5. I think it would be nice to mention that the study provided reference of the distribution of pregnant women’s PROMIS score as a strength of their study.
Typos:
1. Line 155 and 159: “p0.05” -> “p<0.05”
2. Line 183: “TNF-” -> “TNF-α”, “IL-1” -> “IL-1β”
3. Line 187: “(p0.001)” -> “(p<0.001)”

---

## Round 0.2 · accepted · Accept

Thank you for addressing all reviewers' comments. We have decided to accept your manuscript. Please make the changes to these tables.

1. In Table 1, please add indentation to all levels of categorical variables, including Parity, Employment, Educational level, and Hometown; for example, indent "Employed" and "Unemployed" under Employment.

2. In Table 1, add a space between the number and the parenthesis. For example, instead of "30.15(3.97)", please write it as "30.15 (3.97)".

3. In Table 3, please use a bracket for the 95% CI. Instead of "33.504 38.56", please write it as "[33.504, 38.56]". In addition, the round is not consistent throughout Table 3.

4. Please cite the SPSS software and use of PROMIS scores.

Reviewer 1 ·

Basic reporting

no comment

Experimental design

no comment

Validity of the findings

no comment

Additional comments

The authors used a cross-sectional study to present the connections between pro-inflammatory cytokines and prenatal fatigue. By stating the novelty with a related study section, fixing subject-verb agreement issues, and fixing tables and figures, the authors clearly addressed all the comments that have been raised up in the previous round of review. Therefore, the article is recommended for publication in the journal.

Reviewer 2 ·

Basic reporting

The manuscript is much improved. I don't have any further comments.

Experimental design

The manuscript is much improved. I don't have any further comments.

Validity of the findings

The manuscript is much improved. I don't have any further comments.

Additional comments

The manuscript is much improved. I don't have any further comments.